# Structural, Thermal, Rheological, and Morphological Characterization of the Starches of Sweet and Bitter Native Potatoes Grown in the Andean Region

**DOI:** 10.3390/polym15224417

**Published:** 2023-11-16

**Authors:** Olivia Magaly Luque-Vilca, Noe Benjamin Pampa-Quispe, Augusto Pumacahua-Ramos, Silvia Pilco-Quesada, Domingo Jesús Cabel Moscoso, Tania Jakeline Choque-Rivera

**Affiliations:** 1Escuela Profesional de Ingeniería en Industrias Alimentarias, Universidad Nacional de Juliaca, Av Nueva Zelandia 631, Juliaca 21101, Peru; oluque@unaj.edu.pe (O.M.L.-V.); tj.choquer@unaj.edu.pe (T.J.C.-R.); 2Facultad de Ingeniería de Alimentos, Universidad Nacional Intercultural de Quillabamaba, Cusco 08741, Peru; augusto.pumacahua@uniq.edu.pe; 3Facultad de Ingeniería y Arquitectura, Universidad Peruana Unión, km 19 Carretera Central, Ñaña, Lurigancho Lima 15457, Peru; 4Facultad de Ingeniería Ambiental y Sanitaria, Universidad Nacional San Luis Gonzaga, Ica 11004, Peru; jesus.cabel@unica.edu.pe

**Keywords:** sweet potato, bitter potato, starch, thermal, rheology, morphological

## Abstract

This study aimed to extract and characterize the morphological, physicochemical, thermal, and rheological properties of the starches of native potatoes grown in the department of Puno. Among the varieties evaluated were sweet native potato varieties Imilla Negra (*Solanum tuberosum* spp. *Andígena*), Imilla Blanca (*Solanum tuberosum* spp. *Andígena*), Peruanita, Albina or Lomo (*Solanum chaucha*), and Sutamari, and the bitter potatoes Rucki or Luki (*Solanum juzepczukii Buk*), Locka *(Solanum curtilobum*), Piñaza (*Solanum curtilobum*), and Ocucuri (*Sola-num curtilobum*), acquired from the National Institute of Agrarian Innovation (INIA-Puno). The proximal composition, amylose content, and morphological, thermal, and rheological properties that SEM, DSC, and a rheometer determined, respectively, were evaluated, and the data obtained were statistically analyzed using a completely randomized design and then a comparison of means using Tukey’s LSD test. The results show a significant difference in the proximal composition (*p* ≤ 0.05) concerning moisture content, proteins, fat, ash, and carbohydrates. Thus, the amylose content was also determined, ranging from 23.60 ± 0.10 to 30.33 ± 0.15%. The size morphology of the granules is 13.09–47.73 µm; for the thermal and rheological properties of the different varieties of potato starch, it is shown that the gelatinization temperature is in a range of 57 to 62 °C and, for enthalpy, between 3 and 5 J/g.

## 1. Introduction

A high number of varieties of potatoes (*Solanum tuberosum*) natively grown in the Peruvian Andes at higher than 3800 m above sea level [1] are of sweet varieties, subspecies andigena, and bitter varieties characterized by their high content of glycoalkaloids [2]. Sweet potatoes are consumed in different foods, being pleasant to the palate. Ref. [3] indicates that bitter potatoes are not used in nature because of their bitter taste but in preparing freeze-dried and dehydrated potato products called chuño, also known as moraya or tunta, which are highly storable [4]. During this process, they lose the bitter taste, which causes the content of glycoalkaloids [3], and they are used in various typical foods. They contain carbohydrates in the form of starches, proteins, vitamins, dietary fiber, minerals, and carotenoids [5]. Thus, sweet potato starch such as the white Imilla variety has 9.2 ± 0.06 moisture, 0.58 ± 0.01% protein, and 0.58 ± 0.01% fiber, while bitter potatoes such as the Locka variety have 10.2 ± 0.03% moisture, 0.74 ± 0.05% protein, 0.07 ± 0.02% fat, and 0.22 ± 0.01% fiber [6], which vary according to geographical location [7].

Moreover, there is a growing interest in extracting starches from unconventional sources for various applications [8,9,10], so bitter and sweet potatoes can be a source of starches as an alternative to substituting commercial starches [1]. It is known that the techno-functional properties of starches depend on factors such as genotype, environmental conditions, or the region where they are grown [10]; these can be used in obtaining active and intelligent films such as freshness monitors or the development of intelligent packaging biomaterials [11], among other uses.

Characterizing starches from different botanical sources is necessary to determine a specific use in various industries [12]. Starches have semicrystalline and water-insoluble characteristics, whose size and structural morphology vary according to biological origin, and are usually between 1 and 100 µm in diameter, a property that influences physicochemical characteristics and possible industrial use [13,14,15,16]. Morphological characteristics such as shape (round, elongated, elliptical, irregular, and oval), type of surfaces, and size can be determined by scanning electron microscopy (SEM) [12].

Conversely, the rheological properties or paste properties (paste temperature, peak viscosity, breakdown viscosity, and final viscosity) of starches in suspensions (10%) when subjected to heating and cooling ramps interest food engineers and others. This study aims to evaluate the structural, thermal, and rheological morphological characteristics of starches extracted from nine varieties of sweet and bitter native potatoes grown more than 3800 m above sea level in the Peruvian highlands.

## 2. Materials and Methods

### 2.1. Materials

The starches used in the study were from native potatoes from Puno, Peru, and grown in locations higher than 3800 m above sea level. The native sweet potato varieties Imilla Negra (S-IN), Imilla Blanca (S-IB), Peruanita (S-PE), Albina or Lomo (S-AL), and Sutamari (S-SU), and bitter potatoes Locka (B-LO), Piñaza (B-PI), Ocucuri (B-OC), and Ruckii (B-RU) (Figure 1), were purchased from the National Institute of Agrarian Innovation (INIA) (Puno, Peru) and commercial samples from Juliaca. 

### 2.2. Starch Extraction

The starch extraction process was performed using a wet milling technique with a slight modification using the method described by [14]. The potatoes were cut into cubes of approximately 1 m^3^, then immersed in water with sodium metabisulfite (0.20 g/L), and liquefied at 3000× *g* for 10 min with distilled water in a ratio of 3:1 (potato:water). The crushed mass passed through the 60-mesh sieve. The filtrate was centrifuged at 10,000× *g* for 5 min. Once the supernatant was removed, distilled water was added, stirring with a spatula. It was centrifuged at 350 RPM until the starch was shown without residues of other components. The precipitate was placed in metal trays and dried in a forced air stove at 40 °C for 24 h, to finally crush in a mill, 60-mesh sieve, and stored in polypropylene bags for further analysis.

### 2.3. Physical and Physicochemical Properties

International standard methods were used to determine moisture (AOAC—950.46), ash (AOAC—942.05), protein (AOAC—984.13), and fat (AOAC—203.05). The carbohydrates were calculated by difference. All assays were made in triplicate.

Amylose and amylopectin were analyzed using the colorimetric method described by [15]. An iodine solution (mixture of 0.0025 M I2/KI 0.0065 M) was prepared and stored at 4 °C until use. The standard amylose curve (CAS 9005-82-7, Sigma-Aldrich, St. Louis, MO, USA) was prepared at 0%, 10%, 20%, 30%, 40%, and 50% in 15 mL test tubes with lid and 8 mL of dimethyl sulfoxide 90% (*v*/*v*) was added; then, they were mixed vigorously for 2 min by vortex and the tubes were heated in a water bath at 85 °C for 15 min with intermittent mixing. The tubes were allowed to cool for about 45 min at room temperature. Subsequently, the sample was diluted with distilled water in a 25 mL fiola. A total of 1 mL of this solution was taken, and 5 mL of iodine solution was added to it and mixed vigorously. Absorbance was measured at 600 nm. All chemicals used were of analytical grade.

### 2.4. Structural Morphology

Scanning electron microscopy model Prisma E SEM (Thermo Fisher Scientific, Waltham, MA, USA) was used to evaluate the morphology of the granules, based on the method of [16] following the procedure with some modifications with a resolution of 5 nm in high vacuum mode. Also, the starch was spread on the double-sided tape fixed to the electron-conducting carrier. The starches were coated with gold using an E102 ion by sputtering (Hitachi Ltd., Tokyo, Japan). The granules were examined under the following conditions: voltage of 15.0 kV, emission current 100 mA, high vacuum (Pa 10.4), distance from 18.9 to 19.9 mm, and 1000 times increase of 15 thousand times of work.

### 2.5. Thermal and Rheological Properties

The enthalpy of gelatinization and their respective starting temperatures (Ti), peak temperature (TP), and final temperature (TF) were determined by a differential scanning calorimeter (DSC-2500) (TA Instruments, New Castle, DE, USA). Suspensions of starch samples in the ratio of 1:5 (starch: water) were prepared 2 h in advance. In a pure aluminum cell with a lid, a mass of approximately 10 mg was hermetically sealed and heated in the DSC on a ramp of 10 °C/min to 100 °C. The enthalpies and gelatinization temperatures were determined using the TRIOS V5.1.1 software.

The paste behavior of starch dispersions during heating was determined by AACC method 61-02 (2000). The starch cell attachment, external cooling system, and RheoCom-pass™ software 1.30 of the Anton Paar Modular Space MultiDrive Rheometer (Model MCR-702e, Graz, Austria) were used. A mass of 20 g of a 10% starch suspension was placed in the measuring cup. The test conditions were: agitation at 900 rpm for 20 initial s, 160 rpm throughout the experiment, heating and maintenance at 50 °C for 1 min, heating up to 95 °C for 4 min with an increased rate of 11.25 °C/min, constant temperature for 5 min and cooling to 50 °C for 4 min with a rate of 11.25 °C/min, and constant temperature at 50 °C for 2 min. Every 2 s, the equipment recorded the temperature and apparent viscosity, obtaining 480 points. From the viscosity profiles, the peak viscosity (mPa·s), peak time (min), pasting temperature (°C), peak temperature (°C), retention force (mPa·s), breakdown viscosity (mPa·s), final viscosity (mPa·s), total setback viscosity (mPa·s), and trough viscosity (mPa·s) were determined. Smoothed derivative tools of the OriginPro 2022b (64-bit) SR1 software were used by a second-degree equation, with 5 points to the right and left, to analyze the positive or negative velocity peaks (dμ/dt). The kinetics of changes during the heating and cooling process were visualized [17].

## 3. Results

### 3.1. Starch Extraction

The yields (%) of starch for sweet and bitter varieties are shown in Table 1.

Table 1 shows that sweet potatoes had higher yields than bitter potatoes. The yield of both potatoes ranged from 9.63 ± 0.31% to 14.99 ± 0.64% for sweet potato, and 6.52 ± 0.09 to 7.79 ± 0.17 for bitter potato. This showed a significant difference (*p* < 0.05) between extracted starch and yields of the different varieties of native potatoes.

### 3.2. Physical and Physicochemical Properties

Table 2 shows the moisture content between sweet and bitter varieties ranges from 9.29 ± 0.31 to 10.31 ± 0.08%. The content of proteins, fat, and ash values are between a range of 0.526 ± 0.04% and 0.776 ± 0.03%, 0.00 ± 0.01% and 0.06 ± 0.02%, and 0.22 ± 0.01% and 0.317 ± 0.03%, respectively. 

Table 3 shows the results of the amylose and amylopectin content for sweet and bitter potato starches.

The amylose content in both potato varieties ranged from 23.60 ± 0.10 to 30.33 ± 0.15%. The sweet variety Imilla Negra had the highest amylose content of 30.33 ± 0.15%, while the bitter varieties B-RU and B-OC had the lowest amylose content of 23.60 ± 0.10%. 

### 3.3. Morphology and Granule Size

Figure 2 and Figure 3 show SEM micrographs of native potato starches obtained for their morphological properties.

### 3.4. Thermal and Rheological Properties

As observed in Figure 4, starches in bitter potatoes have lower gelatinization starting temperatures than starches in sweet potatoes.

The average gelatinization temperatures and enthalpies of the five sweet potatoes are higher than those of the four bitter potatoes. This shows that there is a difference in their gelatinization properties that is noticeable in the consumption of these potatoes. The bitter ones are more intended for the preparation of chuño and moraya (typical foods of the Incas and currently used in Peruvian cuisine) and the sweet potatoes for direct cooking in various food preparations.

Table 4 shows the temperatures and enthalpy of starch gelatinization from sweet and bitter potatoes. Gelatinization temperatures start from 57.4 to 62.07 °C, ∆T from 9.42 to 14.14 °C, and ∆h from 3.95 to 5.08 J/g. 

### 3.5. Rheological Properties

The rheological properties of the starch paste of the nine potato varieties are shown in Figure 5. The Albina or Lomo variety has the lowest viscosity among all varieties. The varieties B-LO and S-IN have intermediate viscosities, and the remaining varieties have viscosities between 5000 and 6000 mPa·s. The final viscosities at 50 °C all present retrogradation, i.e., an increase in viscosity when decreasing the temperature, being that the variety B-LO presents a more significant viscosity than the peak of viscosity in the stage of gel formation.

Properties of starch paste subjected to heating, isotherm, and cooling are shown in Table 5. The properties of starch paste subjected to heating, isotherm to 95 °C, and cooling to 50 °C show that, for all varieties, the SG presents a range of 64.3 to 68.2 °C for viscosity in a range of 9.2 to 10.3 μ (mPa·s) in a time of 4.3 to 5.4 min. The PT presents a range of 95 to 96 °C for viscosity in a range of 4300 to 5100 μ (mPa·s) in a time of 5.1 to 5.6 min.

Figure 6 presents the data on the gel formation rates using the first derivative of the apparent viscosity (mPa·s) as a function of time (min).

In Figure 6, three different curves of viscosity formation rates can be distinguished throughout the heating and cooling process. The peak gel formation rates (dµ/dt) of the varieties B-OC, B-PI, S-SU, S-IB, S-RU, and S-PE (23.60 to 28.73% amylose) show values between 9000 and 11,000 dµ/dt; for the B-IN and B-LO varieties (24.50 and 30.33% amylose), values of approximately 4000 dµ/dt; and for the S-AL variety (27.77% amylose), a speed of 1000 dµ/dt approximately. According to Juhász and Salgó [14], the high amylose content in starches provides low viscosity during gelatinization. From the information obtained, there would be no correlation between the amylose content and the apparent viscosity during gel formation upon heating. In the present work, the sweet variety S-AL presents low viscosity in all stages of gel formation.

## 4. Discussion

### 4.1. Physical and Physicochemical Properties

The starch extraction yield is lower than that of [18], who reported 16.13% of Ecuadorian native potatoes, which can be explained by the extraction method; this research used a wet extraction, where it was observed during the process lost in the filtering operation that part of the starch remains unrecovered, and also, during the peeling of the potato, due to the depth of the eyes and irregularity of the native potatoes, ref. [19] mentions that tubers of considerable size and regular shape reach yields of up to 16%, while for those of small size and with deep eyes, the yield is 8.51%. Therefore, other extraction methods are recommended for these varieties to improve yield testing.

The differences in yield may be due to several reasons, such as the potato variety maturity of the tuber at the time of extraction [20] because the starch initiates the hydrolysis process after the harvest, and the content is low as the fruit matures. Otherwise, the size of the tuber and starch granules can also have an influence [18]; thus, potato starch of sweet varieties such as Imilla Negra possibly has larger granules than starches of bitter varieties.

The results of the proximal composition are below those reported by [16] of 11.62 ± 0.34% for starch from Hausas potato (*Plectranthus rotundifolius*). Similar results were reported by [6]: 9.2 ± 0.06%, 9.3 ± 0.13%, and 10.2 ± 0.03% for starch from white Imilla, black, and Locka potatoes, respectively. Hence, moisture content influences shelf life during storage [21] and is recommended in flours and starches as being at about 12% moisture, a factor that varies by environmental conditions and storage practices [22].

Generally, starches extracted from native potatoes are low in fat, protein, and ash due to removing other components during starch extraction. Likewise, ref. [13] states that the proximal composition of starches depends on the botanical source, as well as the composition and structure of the polymer, to determine the physicochemical properties of starch.

The difference in amylose content depends on the genotype, so it is observed that potatoes of sweet varieties have a higher content of amylose (from 27.77 to 30.33%) compared to potatoes of bitter varieties (from 23.60 to 25.50%). Peruvian commercial potatoes like Colparina, Huayro, Canchan, and Yungay have 16 to 32% amylose content [16]. Besides environmental conditions, soil types during cultivation and cultural practices can also have an influence [19]. In culinary practice, when cooked, sweet potatoes have a soft and floury texture, so they are widely used in roasting or cooking. Moreover, in the Peruvian highlands, bitter potatoes are used to make chuño and moraya. The higher amylose content of starches in some varieties, such as sweet potatoes, may be due to the granule size because the amylose content of tuber starch is directly related to the size of the granules [23]. The content of amylose and amylopectin plays an important role in influencing the functional properties of starches. Those starches rich in amylose are characterized by their high gelling strength for their usefulness in producing pasta, sweets, and bread, and coating fried products [24].

### 4.2. Structural and Morphological Properties

Regarding the shape and size of the starch granules, it is observed that they have elliptical and spherical shapes, similar to those reported by [15]. Moreover, as reported by [25], starches of commercial potatoes such as Colparina, Huayro, Canchan, and Yungay are grown in Otuzco, La Libertad, and Junin, Peru. Ref. [13] states that elongated and some deformed forms were also observed, whereas size and morphology vary according to the biological origin of the product. The biochemistry of amyloplasts and the physiology of products, as well as environmental factors such as temperature, storage, cultural practices, and the size of starch granules, are related to the amylose content of starch [10]. 

The size of the granules for sweet potato varieties followed normal distribution. The particle size ranged from 13.09 to 47.73 μm, 8.21 to 63.70 μm, 17.36 to 47.32 μm, 15.23 to 47.33 μm, and 18.63 to 46.23 μm for Peruanita, Lomo, Imilla Negra, Imilla Blanca, and Sutamari varieties, respectively. For starches of bitter potato varieties, the range was from 15.84 to 59.88 μm, 12.09 to 91.17 μm, 22.32 to 59.36 μm, and 22.32 to 61.31 μm for Ocucuri, Rucki, Locka, and Piñaza varieties, respectively. The average size of starch granules was 26.779 ± 9.50, 31.86 ± 13.41, 31.52 ± 11.41, 28.52 ± 13.41, 29.30 ± 8.32, 35.15 ± 15.84, 38.29 ± 19.44, 41.29 ± 10.32, and 40.01 ± 8. 23 μm for Peruanita, Lomo, Imilla Negra, Imilla Blanca, Sutamari, Ocucuri, Rucki, Locka, and Piñaza varieties, respectively, and the distribution among medium-sized granules (ranging from 7.5 to 15 µm) was 7.15, 7.72, 8.10, 6. 93, 7.92, 0.15, 9.37, 0.10, 9.70, 5.60, and 4.31% for Peruanita, Lomo, Imilla Negra, Imilla Blanca, Sutamari, Ocucuri, Rucki, Locka, and Piñaza varieties, respectively; for large-sized granules (>15 µm), it was 92. 85, 92.28, 91.9, 93.07, 92.08, 99.85, 90.30, 94.4, and 95.69% for Peruanita, Lomo, Imilla Negra, Imilla Blanca, Sutamari, Ocucuri, Rucki, Locka, and Piñaza varieties, respectively. The size of the granules differs in the same variety, showing a unimodal and bimodal distribution. At the same time, the granular appearance is similar between the different varieties, which coincides with what was stated by [26], which states that there are morphological differences between potato starches compared to other crops from any other botanical source [16]; for example, sizes ranging from 3 to 23 μm (medium and small) were found for Kamu Kamu starch in contrast to large ovoid or cuboid potato starch granules due to differences in the stability of crystalline structures [27], the starch granules in different apple starches ranged from 4.1 to 12.0 µm and were round granules [28], and the Hausa potato starch granule size varied from 3.31 μm to 6.61 μm, having a morphology of some truncated circular shapes [16].

### 4.3. Thermal and Rheological Properties

The conditions for the gelatinization of Native cassava starch had a Tp of 65 °C, ∆T of 20 °C, and ∆h of 13.3 J/g [29]; potato starch native to China had a Tp of 79.04 °C, ∆T of 17.77 °C, and ∆h of 10.34 J/g [30]; Kamu Kamu starch had a Tp 68.7 and ∆h of 16.0 J/g [27], which are values higher than the native potatoes in this study. These results could be related to the fact that native potatoes are easy to cook and gelatinize, and were attributed to differences in granule structure and in the stability of the crystalline structures inside the granule [27]. These thermal properties serve to evaluate the quality of the potato and have a significant impact on food processing [31]. For the different varieties of potato starch, it is shown that the gelatinization temperature is in a range of 57 to 62 °C and, for enthalpy, between 3 and 5 J/g. Ref. [32] indicates that both genotype and environmental conditions like water, temperature, nutritional status, and others significantly affect thermal properties. There are several studies on the effects of environmental conditions on the physical and chemical properties and structural characteristics of postharvest potato starch [7].

Starch granules undergo structural and morphological changes when heated in water, as well as a loss of crystallinity due to the dissociation of amylopectin, inflammation of starch due to water absorption, and amylose leaching to the aqueous phase. These series of changes are generally referred to as starch gelatinization [33]. The starches’ initial gelatinization temperature (To) values were relatively similar to that reported for potato starches native to Puno, Peru, which was 58.1 °C [6]. Starches from four commercial potato varieties ranged from 57.90 to 62.23 °C [25] and were higher than those reported for Sipiera potato starch from Cameroon, with a temperature of 55.22 °C [34]. The peak gelatinization temperature (Tp) was relatively similar to those reported for starch from four commercial potato varieties, with a temperature of 61.18 to 64.85 °C [25] compared to potato starches native to Puno Peru at 61.9 °C [6]. Studies indicate that starches with gelatinization temperatures below 70 °C make it feasible to include them in products such as soft candies, custards, puddings, etc. The final gelatinization temperature (Tc) was similar to that reported for potato starches native to Puno, Peru, which was 68.3 °C [6] compared to starch from four commercial potato varieties from Peru at a range of 65.5 to 68.34 °C; however, it was higher than that reported for Sipiera potato starch from Cameroon, with a value of 65.0 °C [34]. Regarding the enthalpy of gelatinization (ΔH), the range found of 13.76−19.65 J/g was slightly higher than that reported for potato starches native to Puno-Peru with a range of 15.6−15.8 J/g [6] and for starch from four commercial potato varieties with a range of 11.49 to 15.43 J/g [25].

Enthalpy of gelatinization provides a global measure of crystallinity that would indicate a loss of molecular order within the granule during gelatinization [14]. Variations in the gelatinization properties of starches could be attributed to several factors: the morphology, granule size, mineral composition, and molecular structure of the crystalline region of starches [24]. Furthermore, it is worth mentioning that starch gelatinization temperatures vary with the origin, environmental conditions, tuber maturity, and plant age: [35] found a viscosity peak in potato starches close to Ana cultivars with a value of 9254.52 μ [mPa·s] and, for Pirassu, with a value of 9340.38 μ [mPa·s] at temperatures in a range of 66.33–69.95 °C. Ref. [36] found potato starch temperature values at 64.88 °C with a peak viscosity of 9099.6 μ [mPa·s], breakdown viscosity of 6758.72 μ [mPa·s], and final viscosity of 3035.88 μ [mPa·s], and a tendency toward retrogradation with a viscosity of 734.88 μ [mPa·s], which are values that are relatively higher than those found in this study. Ref. [37] analyzed different potato cultivars, and they found potato starch temperatures ranging from 60.4 to 62.4 °C and a maximum viscosity of 4185.72 to 6507.72 μ [mPa·s], which are values that are relatively similar to those found in this study. According to [38], the paste properties of starches are affected by the levels of amylose, lipids, and phosphorus, and the distribution of branch chain lengths of amylopectin. Thus, the viscoelastic properties of starch from different cultivars measured during heating and cooling differed significantly [28].

## 5. Conclusions

The yields obtained range from 6.52 ± 0.09 to 14.99 ± 0.64%. The highest yield variety is Imilla Negra, obtained at 14.99 ± 0.64%, and the lowest yield is 6.52 ± 0.09%, corresponding to the Locka variety. The proximal composition of starches extracted from native potatoes, regarding moisture content, had statistically significant differences between varieties and was from 9.29 ± 0.31 to 10.31 ± 0.08%. The content of proteins, fat, and ash had statistically significant differences, and the values were between a range of 0.526 ± 0.04 and 0.776 ± 0.03%, 0.00 ± 0.01 and 0.06 ± 0.02%, and 0.22 ± 0.01 and 0.317 ± 0.03%, respectively. The amylose content of the nine varieties of native potatoes was from 23.60 ± 0.10 to 30.33 ± 0.15%. The Imilla Negra variety had the highest amylose content of 30.33 ± 0.15%, while the Rucki and Ocucuri varieties had a low amylose content of 23.60 ± 0.10%. The size morphology of the granules followed a normal distribution, with the particle size ranging from 13.09 to 47.73 μm, 8.21 to 63.70 μm, 17.36 to 47.32 μm, 15.23 to 47.33 μm, and 18.63 to 46.23 μm for the Peruanita, Lomo, Imilla Negra, Imilla Blanca, and Sutamari varieties, respectively. In contrast, the starch of the bitter potato varieties presented a range of 15.84 to 59.88, 12.09 to 91.17, 22.32 to 59.36, and 22.32 to 61.31 μm in diameter of the varieties Ocucuri, Rucki, Locka, and Piñaza, respectively. Thermal and rheological properties for the different potato starch varieties show that the gelatinization temperature ranged from 57.2 to 62.3 °C and, for enthalpy, between 3.2 to 5.4 J/g. The Lomo variety had the lowest peak viscosity among all varieties. The Locka and Imilla Negra varieties had intermediate viscosities, and the remaining varieties had viscosities between 5.2 and 6.3 thousand mPa·s. As for the final viscosity at 50 °C, all had retrogradation, being that the Locka variety had a viscosity higher than the peak viscosity in the gel formation stage.

## Figures and Tables

**Figure 1 polymers-15-04417-f001:**
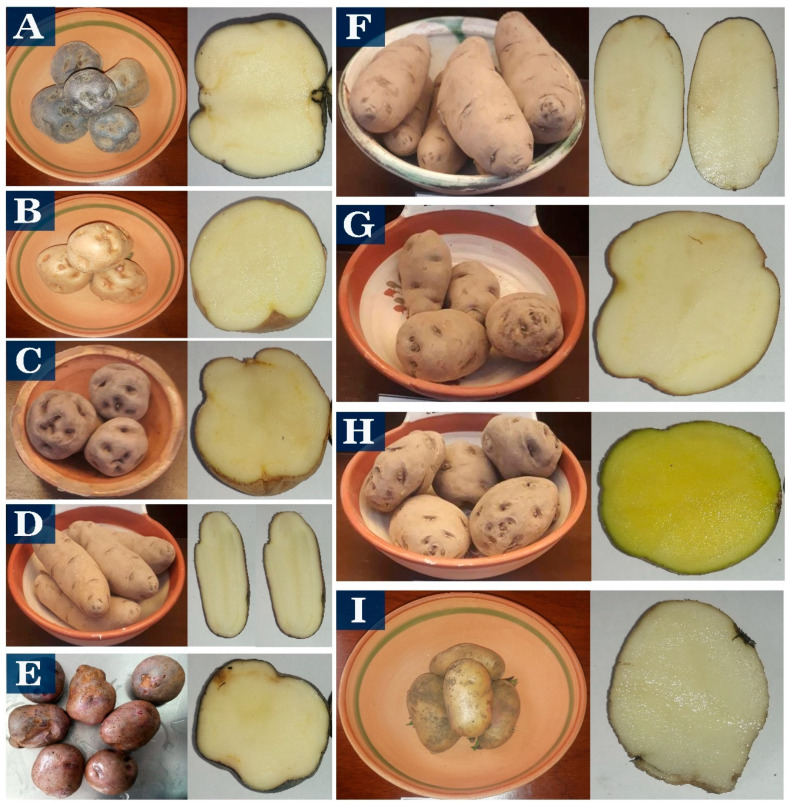
Sweet potato varieties S-IN (**A**), S-IB (**B**), S-PE (**C**), B-LO (**D**), S-SU (**E**) and bitter potato varieties B-LO (**F**), B-PI (**G**), B-OC (**H**), B-RU (**I**).

**Figure 2 polymers-15-04417-f002:**
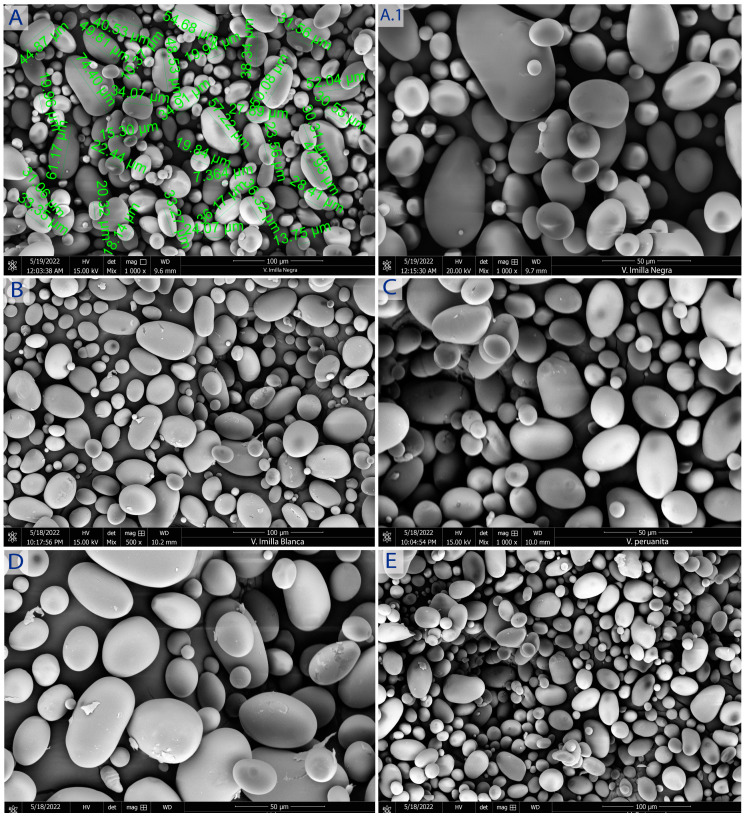
Morphology of sweet potato starches S-IN (**A**,**A.1**), S-IB (**B**), S-PE (**C**), B-LO (**D**), S-SU (**E**). SEM micrographs at 15 kV, 50 and 100 μm.

**Figure 3 polymers-15-04417-f003:**
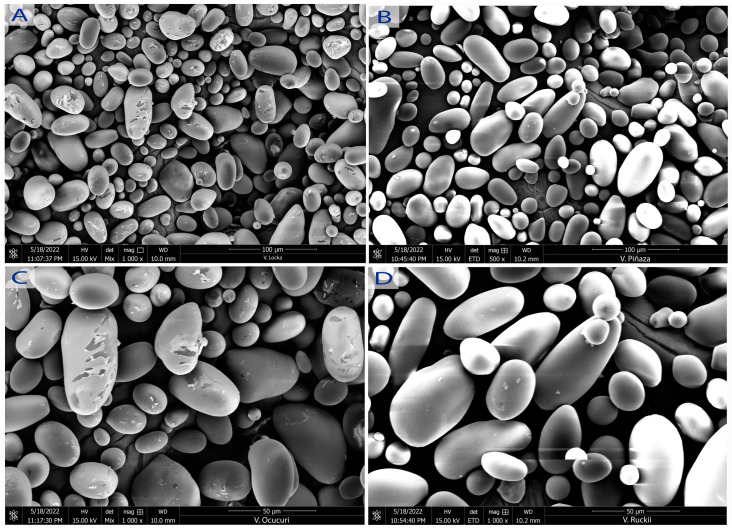
Morphology of starches of bitter potatoes B-LO (**A**), B-PI (**B**), B-OC (**C**), B-RU (**D**). SEM micrographs at 15 kV, 50 and 100 μm.

**Figure 4 polymers-15-04417-f004:**
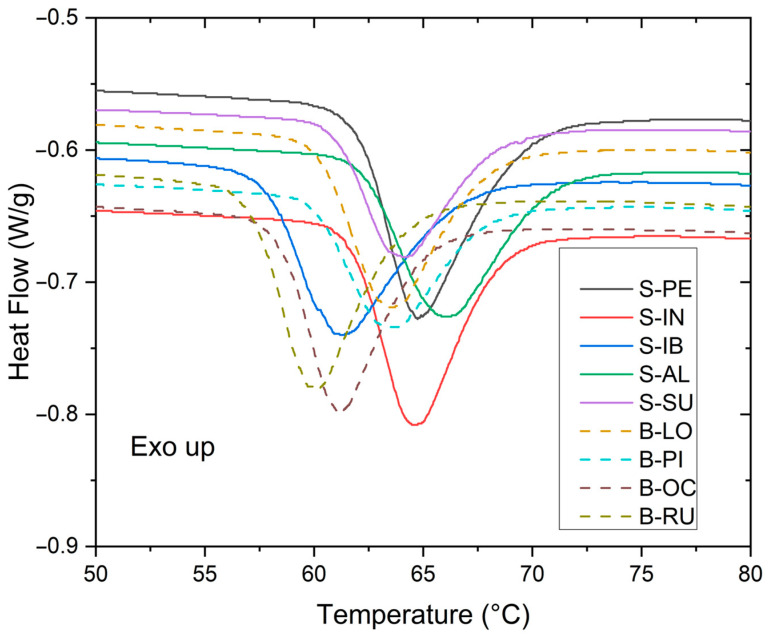
Curves of gelatinization events of sweet and bitter potato starches.

**Figure 5 polymers-15-04417-f005:**
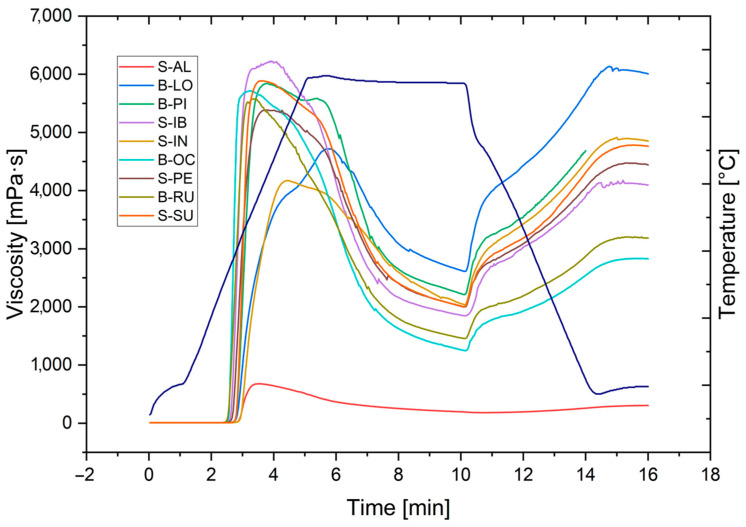
Rheological properties or pasting properties of sweet and bitter potato starches.

**Figure 6 polymers-15-04417-f006:**
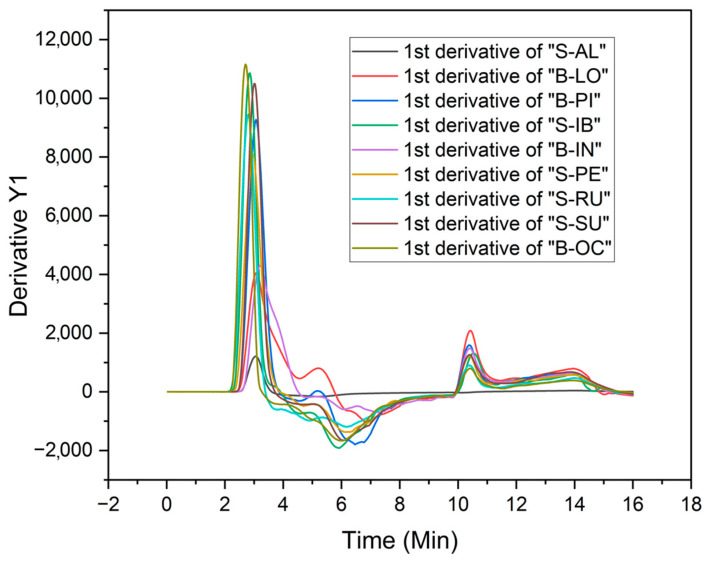
First derivative of the apparent viscosities during heating and cooling of sweet and bitter potato starches.

**Table 1 polymers-15-04417-t001:** Yields of starches.

Native Potato Variety	Mass of Potato (kg)	Mass of Starch (kg)	Shrinkage (kg)	Performance (%)
Sweet				
(S-IN)	10.67 ± 0.17	1.60 ± 0.09	9.07 ± 0.10	14.99 ± 0.64
(S-IB)	10.64 ± 0.03	1.08 ± 0.01	9.56 ± 0.04	10.15 ± 0.12
(S-PE)	10.25 ± 0.38	0.97 ± 0.01	9.26 ± 0.37	9.63 ± 0.31
(S-AL)	10.64 ± 0.32	1.09 ± 0.32	9.56 ± 0.32	10.22 ± 0.32
(S-SU)	10.54 ± 0.31	1.03 ± 0.02	9.51 ± 0.30	9.77 ± 0.13
Bitter				
(B-LO)	10.79 ± 0.12	0.70 ± 0.01	10.09 ± 0.12	6.52 ± 0.09
(B-PI)	11.09 ± 0.20	0.81 ± 0.01	10.28 ± 0.20	7.28 ± 0.16
(B-OC)	11.30 ± 0.35	0.88 ± 0.01	10.42 ± 0.34	7.79 ± 0.17
(B-RU)	10.74 ± 0.13	0.82 ± 0.03	9.92 ± 0.11	7.63 ± 0.16

Values are expressed as mean ± SD (*n* = 3).

**Table 2 polymers-15-04417-t002:** Proximate composition of native potato starches (% dry weight).

Native Potato Varieties	Moisture	Carbohydrates	Protein	Fat	Ash
Sweet					
(S-IN)	9.36 ± 0.08	99.11 ± 0.03	0.56 ± 0.02	0.01 ± 0.01	0.317 ± 0.03
(S-IB)	9.68 ± 0.52	99.15 ± 0.02	0.58 ± 0.03	0.013 ± 0.01	0.25 ± 0.01
(S-PE)	9.56 ± 0.12	99.22 ± 0.05	0.526 ± 0.04	0.01 ± 0.01	0.247 ± 0.02
(S-AL)	9.64 ± 0.23	99.22 ± 0.03	0.53 ± 0.03	0.00 ± 0.01	0.24 ± 0.01
(S-SU)	9.93 ± 0.15	99.21 ± 0.02	0.537 ± 0.02	0.01 ± 0.01	0.24 ± 0.01
Bitter					
(B-LO)	10.31 ± 0.08	98.99 ± 0.07	0.73 ± 0.06	0.04 ± 0.01	0.23 ± 0.02
(B-PI)	10.23 ± 0.01	99.01 ± 0.02	0.67 ± 0.01	0.06 ± 0.02	0.26 ± 0.01
(B-OC)	9.29 ± 0.31	99 ± 0.03	0.776 ± 0.03	0.00 ± 0.01	0.22 ± 0.01
(B-RU)	10.15 ± 0.12	99.10 ± 0.02	0.66 ± 0.01	0.00 ± 0.01	0.23 ± 0.01

Values are expressed as mean ± SD (*n* = 3).

**Table 3 polymers-15-04417-t003:** Amylose and amylopectin contents (%).

Potato Variety	Amylose	Amylopectin
Sweet		
(S-IN)	30.33 ± 0.15	69.67 ± 0.15
(S-IB)	28.20 ± 0.10	71.80 ± 0.10
(S-PE)	28.40 ± 0.10	71.60 ± 0.10
(S-AL)	27.77 ± 0.15	72.23 ± 0.15
(S-SU)	28.73 ± 0.15	71.26 ± 0.15
Bitter		
(B-LO)	24.50 ± 0.30	75.50 ± 0.30
(B-PI)	23.70 ± 0.10	76.30 ± 0.10
(B-OC)	23.60 ± 0.10	76.40 ± 0.10
(B-RU)	23.60 ± 0.10	76.40 ± 0.10

Values are expressed as mean ± SD (*n* = 3).

**Table 4 polymers-15-04417-t004:** Thermal properties of gelatinization temperature and enthalpy of sweet and bitter starches.

Variety	T_i_ (°C)	T_p_ (°C)	T_c_ (°C)	∆T (°C)	∆h (J/g)
Sweet					
(S-IN)	61.77 ± 0.14 ^A^	64.66 ± 0.23 ^AB^	73.02 ± 0.5 ^AB^	11.25 ± 0.77 ^ABC^	4.91 ± 0.96 ^A^
(S-IB)	57.96 ± 0.16 ^CD^	61.3 ± 0.26 ^D^	71.61 ± 1.09 ^BC^	13.65 ± 1.58 ^AB^	3.97 ± 0.63 ^A^
(S-PE)	62.07 ± 0.18 ^A^	65.73 ± 0.33 ^A^	76.2 ± 0.73 ^A^	14.14 ± 0.79 ^A^	4.91 ± 1.11 ^A^
(S-AL)	60.78 ± 0.10 ^B^	63.79 ± 0.27 ^BC^	71.97 ± 0.25 ^BC^	11.19 ± 0.45 ^ABC^	4.05 ± 1.44 ^A^
(S-SU)	61.96 ± 0.15 ^A^	64.56 ± 0.33 ^B^	71.38 ± 0.83 ^BC^	9.42 ± 1.24 ^C^	5.08 ± 0.72 ^A^
Bitter					
(B-LO)	60.06 ± 0.17 ^B^	63.36 ± 0.19 ^C^	72.47 ± 0.17 ^BC^	12.42 ± 0.01 ^ABC^	4.43 ± 0.45 ^A^
(B-PI)	60.37 ± 0.24 ^B^	63.87 ± 0.20 ^BC^	69.82 ± 0.53 ^CD^	9.46 ± 0.47 ^C^	4.34 ± 10 ^A^
(B-OC)	58.59 ± 0.15 ^C^	61.3 ± 0.10 ^D^	69.42 ± 0.98 ^CD^	10.82 ± 1.19 ^BC^	3.95 ± 0.15 ^A^
(B-RU)	57.4 ± 0.16 ^D^	60.33 ± 0.16 ^D^	68.04 ± 0.44 ^D^	10.65 ± 0.49 ^C^	4.32 ± 0.69 ^A^

Starting temperature (T_i_, °C), peak temperature (T_p_, °C), conclusion temperature (T_c_, °C), temperature range (∆T, °C), and the enthalpy of gelatinization (∆h, J/g). Values are expressed as mean ± SD (*n* = 3). Different letters in the same column means indicate significant difference (*p* < 0.05) among samples.

**Table 5 polymers-15-04417-t005:** Properties of starch paste subjected to heating, isotherm, and cooling.

Varieties	Symbol (Unit)	SG	PV	PT	RF	BV	FV
Sweet							
S-IN	T (°C)	67.3 ± 0.1 ^B^	-	95.3 ± 0.2 ^B^	-	-	-
	μ (mPa·s)	9.3 ± 0.4 ^B^	4078.3 ± 75.1 ^E^	4066.3 ± 109.1 ^C^	2034 ± 111.1 ^B^	2144.3 ± 130.2 ^E^	4851 ± 128.7 ^BC^
	t (min)	2.7 ± 0 ^A^	4.5 ± 0.1 ^B^	5 ± 0 ^C^	10.1 ± 0 ^B^	-	-
S-IB	T (°C)	64.9 ± 0.1 ^E^	-	95.4 ± 0.2 ^B^	-	-	-
	μ (mPa·s)	9.6 ± 0.3 ^B^	6158 ± 59.4 ^A^	5604.7 ± 46.7 ^A^	1818.3 ± 48.8 ^D^	4339.3 ± 60.2 ^A^	4070.3 ± 87.1 ^E^
	t (min)	2.4 ± 0 ^DE^	3.7 ± 0.2 ^CD^	5 ± 0 ^C^	10.1 ± 0 ^B^	-	-
S-PE	T (°C)	65.4 ± 0.1 ^D^		95.5 ± 0.2 ^B^	-	-	-
	μ (mPa·s)	9.6 ± 0.5 ^B^	5323.3 ± 65.1 ^C^	4982.7 ± 96.9 ^B^	1943.7 ± 58.3 ^CD^	3380 ± 7.9 ^D^	4456 ± 66.1 ^D^
	t (min)	2.4 ± 0 ^D^	4.1 ± 0.3 ^C^	5 ± 0 ^C^	10.1 ± 0 ^B^		
S-AL	T (°C)	68.1 ± 0 ^A^	-	95.4 ± 0.1 ^B^	-	-	-
	μ (mPa·s)	68.1 ± 0 ^A^	677.9 ± 17.3 ^F^	505.8 ± 7.1 ^D^	181.3 ± 1.6 ^G^	496.6 ± 17.5 ^F^	305.2 ± 3.2 ^H^
	t (min)	2.6 ± 0 ^A^	3.5 ± 0 ^DEF^	5 ± 0 ^C^	10.8 ± 0.1 ^A^	-	-
S-SU	T (°C)	67.3 ± 0.1 ^B^	-	95.4 ± 0 ^B^	-	-	-
	μ (mPa·s)	9.4 ± 0.3 ^B^	5865.3 ± 166 ^AB^	5364.3 ± 134.4 ^A^	2006 ± 9.8 ^BCD^	3859.7 ± 157.3 ^BC^	4640 ± 145.4 ^CD^
	t (min)	2.6 ± 0 ^B^	3.7 ± 0 ^DE^	5 ± 0 ^C^	10.1 ± 0 ^B^	-	-
Bitter							
B-LO	T (°C)	66.8 ± 0.1 ^C^	-	96.1 ± 0.1 ^A^	-	-	-
	μ (mPa·s)	9.8 ± 0.4 ^B^	4721.3 ± 26.3 ^D^	4721.3 ± 26.3 ^B^	2610.3 ± 69.2 ^A^	2111 ± 42.5 ^E^	6007.7 ± 60.4 ^E^
	t (min)	2.5 ± 0 ^C^	5.8 ± 0.1 ^A^	5.8 ± 0.1 ^A^	10.1 ± 0 ^B^	-	-
B-PI	T (°C)	67 ± 0.1 ^C^	-	96 ± 0 ^A^	-	-	-
	μ (mPa·s)	10.2 ± 0.3 ^B^	5735.7 ± 272.4 ^B^	5527.7 ± 207.3 ^A^	2201 ± 48.8 ^B^	3534.3 ± 225 ^CD^	4975.3 ± 19.7 ^B^
	t (min)	2.6 ± 0 ^BC^	3.9 ± 0.2 ^CD^	5.4 ± 0 ^B^	10.1 ± 0 ^B^	-	-
B-OC	T (°C)	64.7 ± 0.2 ^E^		95.4 ± 0.1 ^B^	-	-	-
	μ (mPa·s)	9.2 ± 0.2 ^B^	5647.3 ± 67 ^BC^	4698.3 ± 71.6 ^B^	1233.3 ± 22.9 ^F^	4414 ± 68.8 ^A^	2808.3 ± 17.4 ^G^
	t (min)	2.4 ± 0 ^E^	3.2 ± 0.1 ^F^	5 ± 0 ^C^	10.1 ± 0 ^B^		-
B-RU	T (°C)	63.7 ± 0.2 ^F^	-	95.3 ± 0.1 ^B^	-	-	-
	μ (mPa·s)	9.4 ± 0.8 ^B^	5488.3 ± 226.6 ^BC^	4340.3 ± 105.5 ^C^	1504 ± 144.1 ^E^	3984.7 ± 139.1 ^B^	3264.3 ± 83.7 ^F^
	t (min)	2.3 ± 0 ^F^	3.3 ± 0 ^EF^	5 ± 0 ^C^	10.1 ± 0 ^B^	-	-

Abbreviations: SG = start of gelatinization, PV = peak viscosity, PT = peak temperature, RF = retention force, BV = breakdown viscosity, FV = final viscosity. Different letters in the same column indicate significant difference (*p* < 0.05) among samples.

## Data Availability

The data is original. The data presented in this study are available on request from the corresponding author.

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
