# Peer review of "Structural, Thermal, Rheological, and Morphological Characterization of the Starches of Sweet and Bitter Native Potatoes Grown in the Andean Region"

_polymers, 2023, doi:10.3390/polym15224417_

Round 1
Reviewer 1 Report
Comments and Suggestions for Authors
In this study, the composition, morphology, and thermal and rheological properties of starches from the different potato varieties were comprehensively determined and analyzed. The conception and main content of this manuscript would certainly attract researchers of food science and botany. But the use of language and grammar in this whole manuscript needs to be well polished by native professionals.
1. The exact meaning of this study needs to be introduced and emphasized in the Introduction and Abstract.
2. In the title, for “and rheological”, are the two words in the opposite order?
3. In lines 26-28, “moistened” should be “moisture”, and the two sentences from line26-line28 were repeated.
4. Noun abbreviations in the text only need to appear once, such as SEM.
5. The arrangements of images in figure 1 and figure 2 were in disorder without unified scales, were they in the same magnification? The figures should be rearranged with a clear scale.
6. The statistics in the Results need more comparison and analysis.
Comments on the Quality of English LanguageThe English writing is quite poor with grammatical mistakes. The use of language and grammar in this whole manuscript needs to be well polished by native professionals
Author Response
For research article
|
Response to Reviewer 01 Comments
|
||
|
1. Summary |
|
|
|
Thank you very much for taking the time to review this manuscript. Please find the detailed responses below and the corresponding revisions/corrections highlighted/in track changes in the re-submitted files.
|
||
|
2. Questions for General Evaluation |
Reviewer’s Evaluation |
Response and Revisions |
|
Does the introduction provide sufficient background and include all relevant references? |
Can be improved |
The suggested observation was corrected |
|
Are all the cited references relevant to the research? |
Can be improved |
The suggested observation was corrected |
|
Is the research design appropriate? |
Yes |
|
|
Are the methods adequately described? |
Must be improved |
The suggested observation was corrected |
|
Are the results clearly presented? |
Must be improved |
The suggested observation was corrected |
|
Are the conclusions supported by the results? |
Yes |
|
|
3. Point-by-point response to Comments and Suggestions for Authors |
||
|
Comments from Reviewer No. 1: In this study, the composition, morphology, and thermal and rheological properties of starches from the different potato varieties were comprehensively determined and analyzed. The conception and main content of this manuscript would certainly attract researchers of food science and botany. But the use of language and grammar in this whole manuscript needs to be well polished by native professionals.
|
||
|
Comments 1: The exact meaning of this study needs to be introduced and emphasized in the Introduction and Abstract. |
||
|
Response 1: Thank you for pointing this out, we added more details in the introduction (Section 1, Pag. 1-2, Lines: 40-47, 56-60).
|
||
|
Comments 2: In the title, for “and rheological”, are the two words in the opposite order? |
||
|
Response 2: We appreciate your recommendations re-wrote the title: “Structural, thermal, rheological, and morphological characterization of the starches of sweet and bitter native potatoes grown in the andean region” |
||
|
Comments 3: In lines 26-28, “moistened” should be “moisture”, and the two sentences from line26-line28 were repeated. |
||
|
Response 3: We apologize for the duplicity; we changed the verb form of moistened to moisture and deleted the repeated sentence (Section Abtract, Pag. 1, Lines 27-28). |
||
|
Comments 4: Noun abbreviations in the text only need to appear once, such as SEM. |
||
|
Response 4: We appreciated your recommendations; however, we considered essential mentions for noun abbreviations for more practicality. Besides, we carried out a brief review of other published articles by Polymers using the same technology, e.g., SEM, which they used more than once. Therefore, we used the abbreviations six times in the manuscript (Section Abtract, Pag. 1, Lines 24, Section 1, Pag. 2, Line 62, Section 2.4, Pag. 3 Line 112, Section 3.3, Pag. 7, Line 190, 197). |
||
|
Comments 5: The arrangements of images in figure 1 and figure 2 were in disorder without unified scales, were they in the same magnification? The figures should be rearranged with a clear scale. |
||
|
Response 5: We agree with this comment; we add specifications of each SEM micrograph like scale in 50 and 100 μm (Section 3.3, Pag 7-8, Figure 2 and 3). |
||
|
Comments 6: The statistics in the Results need more comparison and analysis. |
||
|
Response 6: We modified Tables 4 and 5 to improve sample comparisons, indicating significant differences (p < 0.05). Moreover, we added more details in the discussion (Lines 204-213, 238-247, 248- 252, 319-324). The results of the comparison of means by the Tukey test have been added using the letters in the superscript of the averages. That is, both Tables were updated. |
||
|
4. Response to Comments on the Quality of English Language |
||
|
Point 1: The English writing is quite poor with grammatical mistakes. The use of language and grammar in this whole manuscript needs to be well polished by native professionals |
||
|
Response 1: revised with respect to English translation |
||
|
5. Additional clarifications |
||
Reviewer 2 Report
Comments and Suggestions for Authors
This work reports the structural, thermal, morphological, and rheological properties of starches extracted from different sweet and bitter potatoes grown in the area at high altitudes. The topic of this work is interesting. However, several points need to be clarified and corrected as follows:
1. The article title is grammatically incorrect.
2. The authors should provide a picture of the different types of potatoes used in this study in the manuscript so the readers can understand the differences better.
3. The authors mention in the introduction section regarding the differences in chemical compositions and starch properties extracted from the potatoes obtained from different geographical areas. However, the authors fail to pinpoint these differences. The authors must provide comprehensive comparisons with the previous literature's data and sensible explanations. Otherwise, this manuscript does not differ from the nutrition report.
4. No scale bars are found in SEM images (Figures 2-3).
5. In lines 321-323, why do the average particle sizes are given in range? The authors must supplement the particle size measurement and particle size distribution profile as well as S.D. instead.
6. The conclusion section should not be split into many paragraphs.
Comments on the Quality of English LanguageModerate editing of English language is required.
Author Response
|
Response to Reviewer 02 Comments
|
||
|
1. Summary |
|
|
|
Thank you very much for taking the time to review this manuscript. Please find the detailed responses below and the corresponding revisions/corrections highlighted/in track changes in the re-submitted files.
|
||
|
2. Questions for General Evaluation |
Reviewer’s Evaluation |
Response and Revisions |
|
Does the introduction provide sufficient background and include all relevant references? |
Can be improved |
The suggested observation was corrected |
|
Are all the cited references relevant to the research? |
Yes |
|
|
Is the research design appropriate? |
Must be improved |
The suggested observation was corrected |
|
Are the methods adequately described? |
Yes |
|
|
Are the results clearly presented? |
Must be improved |
The suggested observation was corrected |
|
Are the conclusions supported by the results? |
Must be improved |
The suggested observation was corrected |
|
3. Point-by-point response to Comments and Suggestions for Authors |
||
|
Comments from Reviewer No. 2: This work reports the structural, thermal, morphological, and rheological properties of starches extracted from different sweet and bitter potatoes grown in the area at high altitudes. The topic of this work is interesting. However, several points need to be clarified and corrected as follows: |
||
|
Comments 1: The article title is grammatically incorrect. |
||
|
Response 1: We appreciated your comment, We contacted a professional translator to improve the quality of the English Language. However, it is not his specialty in food sciences, so we reviewed the manuscript again to improve more details. |
||
|
Comments 2: The authors should provide a picture of the different types of potatoes used in this study in the manuscript so the readers can understand the differences better. |
||
|
Response 2: We agree with this comment; for that reason, we added Figure 1 (Figure 1. Sweet potato varieties S-IN (A), S-IB (B), S-PE (C), B-LO (D), S-SU (E) and bitter potato varieties, B-LO (F), B-PI (G), B-OC (H), B-RU (I)), (Section 2.1, Pag. 3, Figure 1). |
||
|
Comments 3: The authors mention in the introduction section regarding the differences in chemical compositions and starch properties extracted from the potatoes obtained from different geographical areas. However, the authors fail to pinpoint these differences. The authors must provide comprehensive comparisons with the previous literature's data and sensible explanations. Otherwise, this manuscript does not differ from the nutrition report. |
||
|
Response 3: We agree with this comment. The differences between the different potato varieties that correspond to different geographical growing areas are included and specified, as shown in lines 42-47. |
||
|
Comments 4: No scale bars are found in SEM images (Figures 2-3). |
||
|
Response 4: We agree with this comment; we add specifications of each SEM micrograph like scale in 50 and 100 μm (Section 3.3, Pag 7-8, Figure 2 and 3). |
||
|
Comments 5: In lines 321-323, why do the average particle sizes are given in range? The authors must supplement the particle size measurement and particle size distribution profile as well as S.D. instead. |
||
|
Response 5: Agree. We have, accordingly, modified to emphasize this point. The observation was corrected and the particle size measurement and distribution were supplemented, as well as the standard deviation in the average size, which is shown between lines 301 -314 (Section 4.2, Pag 12) |
||
|
Comments 6: The conclusion section should not be split into many paragraphs. |
||
|
Response 6: We have, accordingly, have done put the paragraphs together. And wrote more details (Section 5, Pag. 14, Lines 380-401). |
||
|
4. Response to Comments on the Quality of English Language |
||
|
Point 1: Moderate editing of English language is required. |
||
|
Response 1: revised with respect to English translation |
||
|
|
||
|
5. Additional clarifications |
||
|
|
||
Reviewer 3 Report
Comments and Suggestions for Authors
Structural, thermal, and rheological morphological characterization of the starches of sweet and bitter native potatoes grown in the Andean region is interesting.
The work presented here is exciting but requires major revisions before it can be accepted for publication. My comments are as follows:
1. ‘Moreover, there is a growing interest in extracting starches from unconventional sources for various applications’-Please add more than one references.
2. ‘Starches have semi crystalline and water-insoluble characteristics, whose size and structural morphology vary according to biological origin, a property that influences physicochemical characteristics and possible industrial use’- Please add more than one references.
3. Morphological, thermal and rheological characterization of starch isolated from New Zealand Kamo Kamo (Cucurbita pepo) fruit (https://doi.org/10.1016/j.carbpol.2006.05.021), Morphological, structural, thermal, and rheological characteristics of starches separated from apples of different cultivars (DOI: 10.1021/jf050923j) works are already reported. Please clarify more details about the novelty of this work.
4. The author needs to pay attention to the format specifications of the figures, such as in Figure 1, gap and spacing between A and B figure should be same. Authors can follow figure 2, as it looks good.
5. The authors need to provide scale bar to Figure 1(A-F) and Figure 2(A-D).
6. The authors need to provide other mechanical data for improvement of the quality of this work such as (i) Temperature sweep rheological data at heating and cooling, (ii) Frequency sweep rheological data.
Author Response
|
Response to Reviewer 03 Comments
|
||
|
1. Summary |
|
|
|
Thank you very much for taking the time to review this manuscript. Please find the detailed responses below and the corresponding revisions/corrections highlighted/in track changes in the re-submitted files.
|
||
|
2. Questions for General Evaluation |
Reviewer’s Evaluation |
Response and Revisions |
|
Does the introduction provide sufficient background and include all relevant references? |
Can be improved |
The suggested observation was corrected |
|
Are all the cited references relevant to the research? |
Can be improved |
The suggested observation was corrected |
|
Is the research design appropriate? |
Yes |
|
|
Are the methods adequately described? |
Yes |
|
|
Are the results clearly presented? |
Can be improved |
The suggested observation was corrected |
|
Are the conclusions supported by the results? |
Yes |
|
|
3. Point-by-point response to Comments and Suggestions for Authors |
||
|
Comments from Reviewer No. 3: Structural, thermal, and rheological morphological characterization of the starches of sweet and bitter native potatoes grown in the Andean region is interesting. The work presented here is exciting but requires major revisions before it can be accepted for publication. My comments are as follows: |
||
|
Comments 1: Moreover, there is a growing interest in extracting starches from unconventional sources for various applications’-Please add more than one references |
||
|
Response 1: We appreciated your comment, we added three references about unconventional sources for various applications (Section 1, Pag. 2, Line 50). |
||
|
Comments 2: ‘Starches have semi crystalline and water-insoluble characteristics, whose size and structural morphology vary according to biological origin, a property that influences physicochemical characteristics and possible industrial use’- Please add more than one references.
|
||
|
Response 2: We agree with this comment. Two more references were added, as shown in line 60. |
||
|
Comments 3: Morphological, thermal and rheological characterization of starch isolated from New Zealand Kamo Kamo (Cucurbita pepo) fruit (https://doi.org/10.1016/j.carbpol.2006.05.021), Morphological, structural, thermal, and rheological characteristics of starches separated from apples of different cultivars (DOI: 10.1021/jf050923j) works are already reported. Please clarify more details about the novelty of this work. |
||
|
Response 3 Agree. We have, accordingly, modified to emphasize this point.The recommended was added, considering the article morphological, thermal and rheological characterization of starch isolated from Kamo Kamo (Cucurbita pepo) fruits from New Zealand (https://doi.org/10.1016/j.carbpol.2006.05.021), which is observed in lines 319 - 321; 329. The morphological, structural, thermal and rheological characteristics of starches separated from apples of different cultivars (DOI: 10.1021/jf050923j), which is observed in lines 331-332; 376-378, were also considered. |
||
|
Comments 4: The author needs to pay attention to the format specifications of the figures, such as in Figure 1, gap and spacing between A and B figure should be same. Authors can follow figure 2, as it looks good. |
||
|
Response 4: We apologize, and We revised and modified figures 2 and 3; more detail in response 5. |
||
|
Comments 5: The authors need to provide scale bar to Figure 1(A-F) and Figure 2(A-D). |
||
|
Response 5: We agree with this comment; we add specifications of each SEM micrograph like scale in 50 and 100 μm. (Section 3.1, Pag 7-8, Figure 2 and 3). |
||
|
Comments 6: The authors need to provide other mechanical data for improvement of the quality of this work such as (i) Temperature sweep rheological data at heating and cooling, (ii) Frequency sweep rheological data. |
||
|
Response 6: Thank you very much for the suggestions. We agree with his comments about adding information on rheology during heating and cooling. Rheological data from heating and cooling temperature scans were performed. Frequency sweep rheological data were not performed. Figure 6 and its respective discussion have been added to it (Section 3.4, Pag 10-11, Lines 232-247) |
||
|
4. Response to Comments on the Quality of English Language |
||
|
Point 1: |
||
|
Response 1: revised with respect to English translation |
||
|
5. Additional clarifications |
||
|
In Section 4.1, we also changed the descriptions of the varieties in line 270 for more clarity. In Section 3.3, we changed the citation of figures 2 and 3 and the title of figures 2 and 3, lines 190-191 and 197-198, respectively. We corrected the numbering of Sections 3.3 – 3.4, Pag. 6 and 11, Lines 188 and 252.
|
||
Round 2
Reviewer 1 Report
Comments and Suggestions for Authors
I agree with the publication of this manuscript.
Reviewer 3 Report
Comments and Suggestions for Authors
Recommendation: Publish as is; no revisions needed.
Comments:
After carefully reading the revised manuscript and point-by-point response to reviewers' comments, I can fully understand the authors' argument and purpose. Thus, I recommend this paper for publication without further modification.